# Modulation of Gut Microbiota by Cacao: Insights from an In Vitro Model

**DOI:** 10.3390/cimb47060414

**Published:** 2025-06-03

**Authors:** Jinshil Kim, Sunil Jung, Gyungcheon Kim, Jinwoo Kim, Bokyung Son, Hakdong Shin

**Affiliations:** 1Department of Food Science & Biotechnology, and Carbohydrate Bioproduct Research Center, Sejong University, Seoul 05006, Republic of Korea; jinshil1130@gmail.com (J.K.); sxcq23@naver.com (S.J.); paulkc12@gmail.com (G.K.); jinwoo3239@gmail.com (J.K.); 2Center for Food Bioconvergence, Seoul National University, Seoul 08826, Republic of Korea; 3Department of Food Biotechnology, Dong-A University, Busan 49315, Republic of Korea; bkson@dau.ac.kr

**Keywords:** gut microbiota, enterotype, cacao, microbial compositions, in vitro model

## Abstract

Natural products play a pivotal role in human health by exerting bioactive effects, including the modulation of the gut microbiome. Cacao, a widely consumed natural product, is rich in polyphenols and dietary fiber, which may influence microbial composition and metabolic functions. However, its effects on the gut microbiota remain poorly understood, particularly regarding inter-individual differences. This study investigated the impact of cacao on gut microbiota using an in vitro fecal incubation model with samples from healthy Korean adults. Our findings classified the gut microbiota of Korean individuals into two distinct enterotypes: *Bacteroides* and *Prevotella*. In the *Bacteroides* enterotype, cacao treatment significantly increased the relative abundance of beneficial bacterial genera, including *Roseburia*, *Lachnospiraceae NK4A136*, *Faecalibacterium,* and *Agathobacter*. Conversely, in the *Prevotella* enterotype, cacao treatment was associated with an increase in the relative abundance of *Prevotella*; however, the small sample size and community shifts during incubation limited the robustness of this observation. Functional predictions based on KEGG pathways further revealed enterotype-specific differences. In the *Bacteroides* enterotype, the cacao-treated group exhibited enhanced pathways associated with starch, sucrose, galactose, and thiamine metabolism, which was not observed in the *Prevotella* enterotype. These findings suggest a potential role for cacao as a gut microbiome modulator, highlighting its possible utility in microbiome-targeted dietary interventions and therapeutic strategies.

## 1. Introduction

Cacao (*Theobroma cacao*) and its products are consumed globally for their nutritional and sensory properties [1]. Cacao is a rich source of proteins, lipids, minerals, starch, and pentosans [2] and is abundant in bioactive compounds, particularly polyphenols, which contribute to its antioxidant activity [2,3]. Numerous studies have highlighted the health benefits of cacao consumption, including reduced risks of cancer, cardiovascular diseases, and inflammatory conditions, as well as modulatory effects on the immune, nervous, and gastrointestinal systems [1]. In addition to these systemic effects, cacao has been shown to influence gut microbiota composition. Some studies have suggested that cacao consumption can selectively enhance or inhibit microbial populations [4,5,6]. For instance, fecal samples from cacao-fed rats demonstrated a reduction in *Bacteroides*, *Clostridium*, and *Staphylococcus* populations [4]. Similarly, a cacao-enriched diet in rats led to decreased *Firmicutes* and *Proteobacteria* populations, alongside increased *Tenericutes* and *Cyanobacteria* [5]. While these findings highlight the potential role of cacao in modulating gut microbiota, most studies have been limited to animal models or investigations of specific cacao components in human trials [4,5,6,7]. However, direct investigations into the effects of whole cacao consumption on the human gut microbiota remain scarce, highlighting the need for further research.

The human gut microbiota plays a crucial role in health, contributing to immune regulation, nutrient metabolism, and defense against pathogens [8]. Based on microbial composition, the gut microbiota is classified into three distinct enterotypes characterized by the predominance of *Bacteroides*, *Prevotella*, or *Ruminococcus* [9]. Among these, *Bacteroides* and *Prevotella* enterotypes are the most prevalent worldwide, as shown by fecal microbiota analyses of healthy individuals across five continents [10]. Diet is a key determinant of enterotypes. A protein- and animal-fat-rich diet is associated with the *Bacteroides* enterotype, whereas a carbohydrate- and fiber-rich diet favors the *Prevotella* enterotype [11,12,13]. Given the distinct metabolic capabilities and dietary responses of each enterotype [14], it is essential to evaluate the effects of food in an enterotype-specific context. However, no study has investigated the impact of cacao consumption on the gut microbiota based on enterotype classification, highlighting a critical gap in the literature.

In vitro models that simulate the human gastrointestinal environment are widely used to study the gut microbiota, providing cost-effective, rapid, and high-throughput alternatives to in vivo models [15,16]. These models have been successfully employed to compare short-chain fatty acid production in response to various dietary fibers [17]. In this study, we investigated the impact of cacao on gut microbiota using an in vitro fecal incubation model. Fecal samples from healthy Korean volunteers were analyzed to assess cacao-induced changes in the microbial composition, structure, and predicted metabolic function. Furthermore, we examined whether the microbiota response to cacao varied according to enterotype, providing insights into the potential for personalized dietary interventions tailored to gut microbiota profiles.

## 2. Materials and Methods

### 2.1. Participants Information

This study recruited 48 healthy Korean adults (25 male and 23 female participants) aged 20 to 40 years. Written informed consent was obtained from all participants prior to the sample collection. Data on participants’ age, sex, and body mass index (BMI) were recorded, along with self-reported medical histories, including recent antibiotic therapy (within the past six months) and any diagnosis of inflammatory bowel syndrome (IBS) or inflammatory bowel disease (IBD). The study protocol was approved by the Institutional Review Board (IRB) of Sejong University (IRB No. SJU-BR-E-2020-025).

### 2.2. Preparation of Cacao Products and In Vitro Fecal Incubation

Cacao powder (The Hershey Company, Hershey, PA, USA) was diluted to a 10% (*v*/*v*) solution in 1× phosphate-buffered saline (PBS), following a previously established method [18]. The in vitro fecal incubation model was prepared using a basal medium designed to simulate intestinal conditions, as previously described [19]. Fresh fecal samples (5–10 g per participant) were collected in sterile containers and processed within 30 min of their collection. The samples were diluted at a 1:3 ratio with basal medium, homogenized, and filtered through a sterile nylon mesh (985 µm) [18,19] under anaerobic conditions (5% CO_2_, 5% H_2_ 5%, 90% N_2_). The homogenized fecal samples were adjusted to a final concentration of 2% (0.2 g/mL) and loaded into sterile 96-deep well plates. The diluted cacao solution was added to the fecal samples at a final concentration of 1% (0.1 g/mL). The fecal-cacao mixtures were incubated for 24 h at 37 °C with shaking at 150 rpm in an anaerobic chamber to allow fermentation. Following incubation, the samples were stored at −80 °C until further analysis.

### 2.3. 16S rRNA Gene Amplicon Sequencing

16S rRNA gene sequencing was performed according to the Earth Microbiome Project (EMP) protocols [20]. DNA extraction was performed using the DNeasy PowerSoil Kit (Qiagen, Hilden, Germany) according to the manufacturer’s instructions. Fecal samples were homogenized using a TissueLyser II (Qiagen Retsch GmbH, Hanover, Germany) at 30 Hz for 10 min to ensure efficient disruption of microbial cells. The V4 region of the bacterial 16S rRNA gene was amplified using 515F (AATGATACGGCGACCACCGAGATCTACACGCT) and 806R (CAAGCAGAAGACGGCATACGAGAT) primers with barcode sequence [21]. PCR mixtures consist of 10 μL of Gold hot-start Taq PCR master mix (Bioneer, Daejeon, Korea), 1 μL of each forward and reverse primer (5 μM), 1 μL of the template, and PCR-grade water (Sigma-Aldrich, St. Louis, MO, USA) to a final volume of 25 μL. The cycling conditions were as follows: initial denaturation step at 94 °C for 3 min, 35 cycles of denaturation at 94 °C for 45 s, annealing at 50 °C for 60 s, extension at 72 °C for 90 s, and final extension cycle for 10 min at 72 °C. The PCR products were quantified using the Quant-iT PicoGreen dsDNA Assay Kit (Invitrogen, Carlsbad, CA, USA) and normalized to a uniform concentration. Sequencing was performed on an Illumina MiSeq platform using the 2 × 300 v3 Kit. Primer sequences were trimmed from raw reads, and any read pairs lacking either forward or reverse sequences were removed.

### 2.4. Gut Microbiota Data Analysis

The gut microbiota data were analyzed as described in previous studies [22,23]. Microbial diversity and community composition were analyzed using the QIIME 2 software pipeline (v. 2022.2) [24]. Demultiplexing and quality filtering were performed using the q2-demux plugin, and denoising was conducted using DADA2 [25]. Taxonomic classification was assigned based on SILVA (v.138). Alpha-diversity metrics (Faith’s Phylogenetic Diversity (PD), observed features, and Shannon entropy) were measured using the non-parametric Kruskal−Wallis test. Beta diversity (weighted and unweighted UniFrac) metrics were used to assess differences between groups via permutational multivariate analysis of variance (PERMANOVA) using q2-diversity [26]. Linear discriminant effect size (LEfSe) analysis was carried out to detect significant differences in bacterial composition (linear discrimination analysis (LDA) score > 3.0) [27]. For species-level prediction analysis, which enables the identification of specific amplicon sequence variants (ASVs) at the species level, the Basic Local Alignment Search Tool (BLAST; v2.12.0) was employed. The phylogenetic investigation of communities by reconstruction of unobserved states (PICRUSt2) was used based on the Kyoto Encyclopedia of Genes and Genomes (KEGG) ortholog classification to predict the metabolic function of the metagenomes from the 16S rRNA gene dataset [28]. To identify gut microbiota enterotypes, baseline fecal samples were analyzed at the genus level. Clustering was conducted using the Partitioning Around Medoids (PAM) method based on Jensen–Shannon divergence (JSD) as the distance metric within the R environment [9]. The optimal number of clusters and cluster robustness were determined by evaluating the Calinski–Harabasz (CH) index and silhouette analysis [29].

### 2.5. Metabolite Profiling of Cacao Powder

Metabolites derived from cacao powder were analyzed using a Vanquish U-HPLC system (Thermo Fisher Scientific, Waltham, MA, USA) coupled to an Orbitrap Exploris 120 mass spectrometer (Thermo Fisher Scientific, Waltham, MA, USA). Cacao powder (10 mg) was extracted with 1 mL of 80% methanol by vortexing, followed by centrifugation at 4 °C and 13,000× *g* for 15 min. The resulting supernatant was analyzed by U-HPLC-MS/MS using a C18 column (ACQUITY UPLC BEH C18, 2.1 × 100 mm, 1.7 μm; Waters Corp., Milford, MA, USA) maintained at 45 °C. The mobile phase consisted of water (A) and acetonitrile (B), both containing 0.1% formic acid. Mass spectrometry was performed in the positive and negative ionization modes with a scan range of 55–800 *m*/*z*. Raw data were processed using Xcalibur (version 4.6) and Compound Discoverer (version 3.3). Blank samples were prepared using 80% methanol, and blank subtraction was performed to eliminate background signals. Metabolite identification was performed by matching *m*/*z* values and MS/MS spectra against the mzCloud (accessed in April 2025), ChemSpider (accessed in April 2025), and HMDB (version 5.0) databases.

## 3. Results and Discussion

### 3.1. Comparison of Gut Microbial Composition Between the Two Enterotypes

In vitro fermentation models are essential tools for investigating the effects of specific foods or food-derived components on the gut microbiota [30,31]. In this study, we utilized an in vitro fecal incubation model to evaluate the impact of cacao on gut microbiota.

Enterotypes, primarily dominated by the genera *Bacteroides*, *Prevotella*, and *Ruminococcus*, have been proposed as a way to differentiate individual gut microbiota [9], with *Bacteroides* and *Prevotella* enterotypes being the most frequently observed [32]. A previous study has also shown a correlation between dietary patterns and enterotypes [33], with food intake affecting gut microbiota composition in an enterotype-specific manner. However, to our knowledge, no study has reported how cacao consumption modulates gut microbial composition according to enterotype classification. In this study, we aimed to investigate the enterotype-dependent responses of the gut microbiota to cacao treatment. Microbiota information from fecal samples collected before incubation (0 h) was used for enterotyping, and post-incubation data (24 h) were analyzed to investigate the effect of cacao treatment on the gut microbiota composition.

Enterotype analysis of the initial fecal samples revealed two distinct groups corresponding to the *Bacteroides* (*n* = 39) and *Prevotella* (*n* = 9) enterotypes (Figure 1A and Appendix A). The clustering accuracy was moderate, with a silhouette index value of 0.20 [34] (Figure 1B). This is consistent with previous findings that identified *Bacteroides* and *Prevotella* as the dominant enterotypes in the Korean population [12,33]. Two samples (Sub.17 and Sub.19) were excluded from further analysis due to inconsistent *Prevotella* ratios, which could hinder accurate classification (Appendix A).

At the genus level, we observed significant differences in microbial composition between the two enterotypes, as shown by the top 30 most abundant taxa (Figure 1C). The *Bacteroides* enterotype was predominantly characterized by *Bacteroides*, *Lachnospiraceae*, *Lachnoclostridium*, *Blautia*, *Fusicatenibacter*, and *Bifidobacterium*, whereas the *Prevotella* enterotype was dominated by *Megasphaera*, *Succinivibrio*, and *Prevotella* (LDA > 3.0 for bacterial features in proportions > 1%; Figure 1D).

### 3.2. Differences in Gut Microbiota According to Cacao Treatment

To determine the effect of cacao on the gut microbiota, we compared the non-treated (control) and cacao-treated (cacao) groups using enriched samples after 24 h of incubation. No statistically significant differences were observed in the alpha diversity metrics (Faith’s PD, observed features, and Shannon Entropy) between the control and cacao groups (Figure 2A–C). Similarly, no significant difference was observed in beta diversity based on unweighted UniFrac distances between the control and cacao groups (Figure 2D). However, beta diversity based on weighted intra-group distances was significantly higher in the cacao group than in the control group (Figure 2E).

Microbial composition also differed significantly between the two groups (Figure 2F,G). In the control group, *Parabacteroides*, *Lachnospiraceae* UCG 010, *Alistipes*, and *Dorea* were overrepresented compared to the cacao group (LDA > 3.0 for bacterial features in proportions > 1%; Figure 2F,G). Previous studies have suggested that these genera may have both beneficial and pathogenic effects on human health. *Parabacteroides*, a Gram-negative bacterium found in the intestinal tract, has been linked to potential pathogenic effects on gut wall integrity [35]. Although *Dorea* is part of a healthy gut microbiota, it is also abundant in patients with inflammatory bowel diseases, such as Crohn’s disease [36]. *Alistipes* has been associated with both protective and harmful effects, depending on the disease context. While it has been linked to beneficial roles in some disorders, other studies suggest that it may contribute to colorectal cancer, mood disorders, and cardiovascular diseases [37,38,39].

In contrast, the cacao group showed a significant increase in the relative abundance of *Bifidobacterium*, *Roseburia*, *Lachnospiraceae* NK4A136 group, *Faecalibacterium*, and *Agathobacter* compared to the control group (LDA > 3.0 for bacterial features in proportions > 1%; Figure 2F,G). *Bifidobacterium*, a well-known probiotic, plays a crucial role in gastrointestinal health by alleviating symptoms such as constipation and diarrhea [40,41]. Previous studies have reported increased levels of *Bifidobacterium* in cacao- or its ingredient-treated groups compared to non-treated groups [42,43].

To better understand the components responsible for these effects, we conducted a metabolomic analysis of the cacao powder used in this study. The analysis revealed a wide range of bioactive compounds, including amino acids (e.g., valine, alanine, aspartic acid, glutamic acid, and arginine), short-chain fatty acid (SCFA) precursors (e.g., citric acid, fumaric acid, and malic acid), and various polyphenols and flavonoids, such as epicatechin, isoquercitrin, procyanidin B2, eriodictyol, and vanillin (Appendix A).

Notably, the high polyphenol content of cacao has been shown to promote the growth of *Bifidobacterium* in the gut microbiota. For instance, polyphenols derived from pomegranates have been shown to specifically enhance the growth of *Bifidobacterium* spp. [44]. Similarly, wine polyphenols have been linked to increased *bifidobacteria* levels in rats [45]. *Faecalibacterium*, one of the most abundant and essential commensal bacteria in the human gut, supports host health by producing anti-inflammatory metabolites and contributing to energy metabolism [46]. Additionally, the abundance of SCFA-producing bacteria, including *Roseburia*, *Lachnospiraceae*, and *Agathobacter*, was higher in the cacao group than in the control group (Figure 2F,G). SCFAs play crucial roles in intestinal epithelial cell function, differentiation, and expansion of regulatory T (Treg) cells and exhibit potent anti-inflammatory properties [47,48].

Taken together, these findings suggest that cacao consumption may have beneficial effects on human gut health by promoting the growth of beneficial bacteria and reducing the abundance of potentially pathogenic bacteria in the gut microbiota.

### 3.3. Effect of Cacao on Gut Microbiota Diversity and Structure According to Enterotype

To determine the effect of cacao on gut microbial diversity according to enterotype, we compared the control and cacao groups for each enterotype. Bacterial alpha diversity did not differ significantly between the control and cacao groups in either of the enterotypes. Similarly, in the weighted UniFrac distance-based PCoA analysis, no distinct clustering pattern was observed between the control and cacao groups in the *Bacteroides* enterotype. In contrast, a clustering trend was observed for the *Prevotella* enterotype (*p* = 0.084) (Figure 3A,B). This trend was further supported by inter-group distance analysis, which showed significantly greater distances between groups in the *Prevotella* enterotype than in the *Bacteroides* enterotype (Figure 3C).

We then compared the microbial composition between the control and cacao groups within each enterotype. In the *Bacteroides* enterotype, cacao treatment significantly increased the relative abundance of *Lachnospiraceae* NK4A136 group, *Faecalibacterium*, and *Agathobacter*, while it decreased that of *Lachnospiraceae* UCG 010 (Figure 3D,E). In the *Prevotella* enterotype, cacao treatment was associated with a reduction in the relative abundance of *Lachnoclostridium, Bilophila*, and *Dorea* and an increase in the relative abundance of *Prevotella* (Figure 3D,E and Appendix A). These results suggest that cacao contributes to the maintenance of *Prevotella* populations, particularly in *Prevotella*-dominant microbiota in our dataset. For future studies, incorporating shorter incubation periods (e.g., 6 or 12 h) or in vivo validation may provide a more accurate reflection of microbiota dynamics in response to cacao treatment while minimizing compositional shifts introduced by prolonged in vitro incubation.

Moreover, we further examined these taxa at the species level using the BLAST analysis of DADA2-derived ASVs, identifying *Prevotella copri*, a common *Prevotella* species in the human gut [49], as the most significantly increased species following cacao treatment in all fecal samples classified as the *Prevotella* enterotype (Appendix A). This finding suggests that *P. copri* is particularly responsive to cacao (Appendix A).

As mentioned earlier, a higher relative abundance of *Bifidobacterium* was observed in the cacao group than in the control group (Figure 2G). In particular, the relative abundance of *Bifidobacterium* species, including *Bifidobacterium faecale* and *Bifidobacterium longum* subsp. *longum*, and *Bifidobacterium pseudocatenulatum*, increased following cacao treatment (Appendix A); however, this increase was not statistically significant (Figure 3E). Enterotype-specific analysis revealed that *B. faecalis* increased in the *Bacteroides* enterotype, while both *B. faecalis* and *B. longum* subsp. *longum* were elevated in the *Prevotella* enterotype following cacao treatment (Appendix A). This trend suggests that cacao may promote the growth of beneficial *Bifidobacterium* species, which are known to support gut health. Further studies are required to confirm these effects in vivo and to clarify individual variability.

### 3.4. Predictive KEGG Functional Profiling According to Cacao Treatment

To analyze the functional profiles associated with cacao treatment, we used PICRUSt2 to predict the KEGG functional pathways. In the cacao group, a higher proportion of bacterial genes were associated with carbohydrate metabolism pathways, including starch, sucrose, and galactose metabolism, as well as thiamine and riboflavin metabolism. Conversely, a lower proportion of genes was observed for glycine/serine/threonine/tryptophan metabolism, valine/leucine/isoleucine degradation, tropane/piperidine/pyridine/alkaloid biosynthesis, propanoate/butanoate metabolism, methane metabolism, lipoic acid metabolism, taurine/hypotaurine metabolism, and aminobenzoate/nitrotoluene degradation (LDA > 2.0; Table 1) in the cacao group. Notably, *Bifidobacterium*, which was more abundant in the cacao group, can produce thiamine pyrophosphate (TPP) [50], which aligns with the observed overrepresentation of thiamine metabolism genes in the cacao group (Table 1).

KEGG functional profiling further revealed that in the *Bacteroides* enterotype, the cacao group exhibited a higher representation of genes related to starch, sucrose, and galactose metabolism, secondary and primary bile acid biosynthesis, and thiamine metabolism (LDA > 2.0, Table 1). In contrast, the control group of the *Bacteroides* enterotype showed a higher representation of genes associated with glycine/serine/threonine metabolism, valine/leucine/isoleucine degradation, tropane/piperidine/pyridine/alkaloid biosynthesis, propanoate/butanoate metabolism, lipoic acid metabolism, taurine/hypotaurine metabolism, and aminobenzoate degradation. *Bacteroides* species in the gut are known for their abundance of genes involved in carbohydrate-active enzymes and vitamin/cofactor metabolism [51], supporting these findings.

In the *Prevotella* enterotype, cacao treatment was associated with an increased representation of pantothenate/CoA biosynthesis and a decreased representation of methane metabolism (LDA > 2.0, Table 1). *P. copri*, a species that showed increased relative abundance following cacao treatment, possesses a biosynthesis pathway for vitamin B5 (pantothenate) [50], suggesting a potential link between its enrichment and increased production of vitamin B5. However, further studies, particularly those using metabolomics, are required to validate these predicted functional differences.

## 4. Conclusions

Our findings suggest that cacao can modulate the composition and functional profile of the human gut microbiota. Using an in vitro fecal incubation model with samples from 48 healthy Korean adults, we observed significant shifts in microbial community structure and predicted metabolic pathways following cacao treatment. These alterations included an increase in health-associated bacterial genera and enhancement of pathways related to carbohydrate and vitamin metabolism. While some variation in response appeared to be associated with enterotype, this observation should be interpreted with caution due to several limitations of this study. First, the number of individuals classified into the *Prevotella* enterotype was relatively small (*n* = 9), which may limit the generalizability of the findings related to this group. Second, the in vitro fecal incubation model, while useful for screening, does not fully recapitulate host-related factors, such as digestion, absorption, metabolism, and host–microbe immune interactions. In addition, a shift toward *Bacteroides* dominance was observed in most samples after 24 h of incubation, including those initially enriched with *Prevotella*, indicating that the in vitro system may not fully preserve the baseline microbial community structures. While this study used whole cacao powder in an in vitro fecal model, we acknowledge that digestion may alter its components in vivo. Further, in vivo studies are required to validate these results and gain deeper insights into the impact of cacao on the gut microbiome.

## Figures and Tables

**Figure 1 cimb-47-00414-f001:**
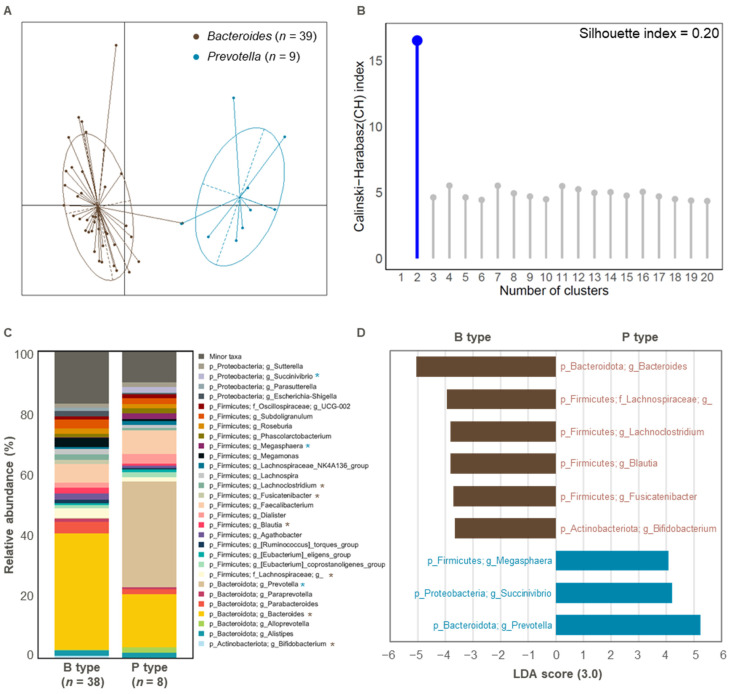
The gut enterotype of initial fecal samples and microbial composition according to enterotype. (**A**) Enterotypes identified in 48 initial fecal samples. The clustering results were visualized using PCoA. (**B**) The optimal number of clusters was obtained using the CH and Silhouette indices (*n* = 48). (**C**) Bacterial taxonomy at the phylum and genus levels (top 30 bacterial taxonomies) of *Bacteroides* (*n* = 38) and *Prevotella* (*n* = 8) types. Significant differences are marked with an asterisk (*, *p* < 0.05). Significant increases in B type or P type are indicated in brown or blue, respectively. (**D**) Relative abundance of overrepresented bacterial taxa (LDA > 3.0) in *Bacteroides* (*n* = 38; brown) and *Prevotella* (*n* = 8; blue) types. B type, *Bacteroides* type; P type, *Prevotella* type.

**Figure 2 cimb-47-00414-f002:**
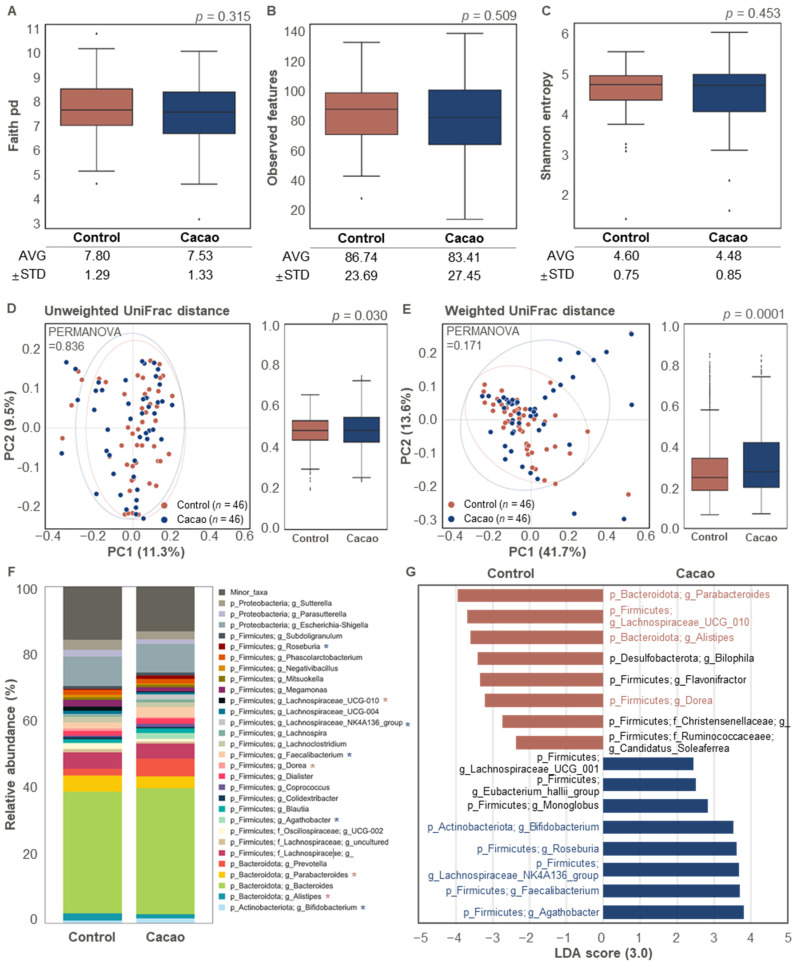
Differences in bacterial structure in the gut microbiota according to cacao treatment (*n* = 46). (**A**–**C**) Comparison of alpha diversity indices between the control and cacao groups. Distribution of (**A**) Faith’s phylogenetic diversity (PD), (**B**) observed features, and (**C**) Shannon entropy was used to evaluate alpha diversity in the gut microbiota. (**D**,**E**) Comparison of beta diversity indices between the control and cacao groups. (**D**) Unweighted and (**E**) weighted UniFrac distances were used to evaluate beta diversity. (**F**) Bacterial taxonomy at the phylum and genus levels (top 30 bacterial taxonomies) in the two groups. Significant differences between the control and cacao groups are marked with an asterisk (*, *p* < 0.05). A significant increase in the control or cacao groups is indicated in brown or blue, respectively. (**G**) Relative abundance of overrepresented bacterial taxa (LDA > 3.0) in the control and cacao groups.

**Figure 3 cimb-47-00414-f003:**
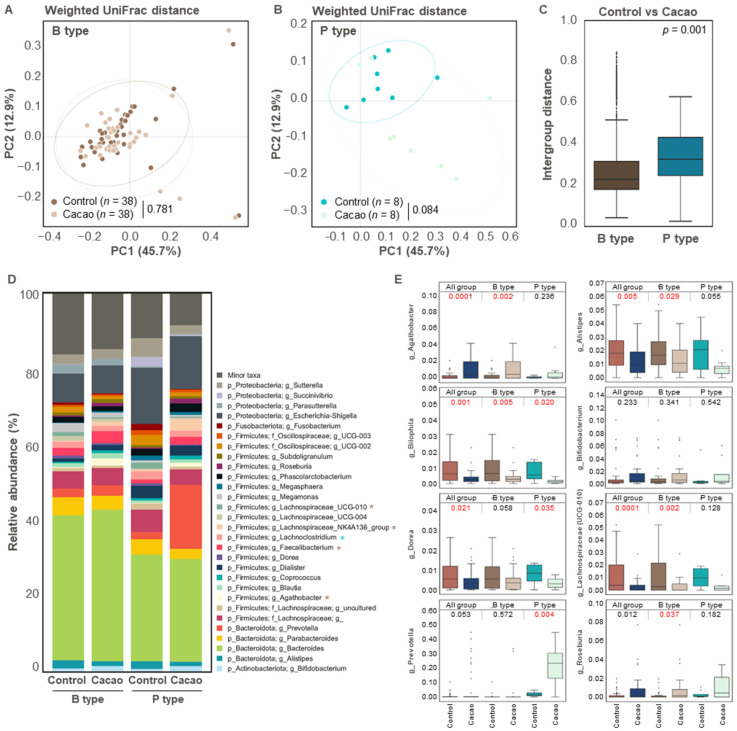
Differences in the gut microbiota according to enterotype (*n* = 46). (**A**–**C**) Comparison of beta diversity indices between the control and cacao groups according to enterotype. Principal Coordinate Analysis (PCoA) plot based on weighted UniFrac distance of two enterotypes, (**A**) *Bacteroides* and (**B**) *Prevotella* types, and (**C**) intergroup distance were used to evaluate beta diversity in gut microbiota between the control and cacao groups. (**D**) Bacterial taxonomy at the phylum and genus levels (top 30 bacterial taxonomies) of the two groups in *the Bacteroides* and *Prevotella* types. Significant differences between the control and cacao groups are marked with an asterisk (*, *p* < 0.05). Significant changes in B type or P type are indicated in brown or blue, respectively. (**E**) Relative abundance of *Agathobacter*, *Alistipes*, *Bilophila*, *Bifidobacterium, Dorea*, *Lachnospiraceae* (UCG-010), *Prevotella*, and *Roseburia* in enriched fecal samples according to cacao treatment. B type, *Bacteroides* type; P type, *Prevotella* type.

**Table 1 cimb-47-00414-t001:** Predictive KEGG functional profiling of the control and cacao groups.

LEVEL2	LEVEL3	All Group (*n* = 46)			B Type (*n* = 38)			P Type (*n* = 8)		
		Control	*p* Value	Cacao	Control	*p* Value	Cacao	Control	*p* Value	Cacao
Amino acid metabolism	Glycine serine and threonine metabolism	2.14	0.001	-	2.12	0.001	-	-	-	-
	Tryptophan metabolism	2.12	0.044	-	-	-	-	-	-	-
	Valine leucine and isoleucine degradation	2.27	0.006	-	2.19	0.018	-	-	-	-
Biosynthesis of other secondary metabolites	Tropane, piperidine, and pyridine alkaloid biosynthesis	2.58	0.006	-	2.59	0.003	-	-	-	-
Carbohydrate metabolism	Starch and sucrose metabolism	-	0.000	2.43	-	0.000	2.38	**-**	**-**	**-**
	Galactose metabolism	-	0.043	2.50	-	0.042	2.47	**-**	**-**	**-**
	Propanoate metabolism	2.31	0.009	-	2.27	0.027	-	-	-	-
	Citrate cycle (TCAcycle)	2.32	0.020	-	2.32	0.038	-	-	-	-
	Butanoate metabolism	2.26	0.002	-	2.26	0.005	-	-	-	-
Energy metabolism	Nitrogen metabolism	2.15	0.033	-	-	-	-	-	-	-
	Carbon-fixation pathways in prokaryotes	2.49	0.019	-	-	-	-	-	-	-
	Methane metabolism	2.13	0.000	-	-	-	-	2.66	0.019	-
Lipid metabolism	Secondary bile acid biosynthesis	-	-	-	-	0.044	2.94	-	-	-
	Primary bile acid biosynthesis	-	-	-	-	0.044	2.33	-	-	-
Metabolism of cofactors and vitamins	Thiamine metabolism	-	0.004	2.53	-	0.004	2.54	**-**	**-**	**-**
	Riboflavin metabolism	-	0.037	2.14	-	-	-	-	-	-
	Lipoic acid metabolism	2.73	0.012	-	2.62	0.032	-	-	-	-
	Pantothenate and CoA biosynthesis	-	-	-	-	-	-	-	0.031	2.74
Metabolism of other amino acids	Taurine and hypotaurine metabolism	2.45	0.014	-	2.46	0.037	-	-	-	-
Xenobiotics biodegradation and metabolism	Aminobenzoate degradation	2.10	0.002	-	2.06	0.007	-	-	-	-
	Nitrotoluene degradation	2.31	0.044	-	-	-	-	-	-	-

## Data Availability

All amplicon sequence data and metadata have been made public through the EMP data portal (Qiita, https://qiita.ucsd.edu; study ID: 15919) accessed on 18 March 2025.

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
