# Peer review of "Modulation of Gut Microbiota by Cacao: Insights from an In Vitro Model"

_cimb, 2025, doi:10.3390/cimb47060414_

Round 1
Reviewer 1 Report
Comments and Suggestions for Authors
The manuscript "Enterotype-Specific Modulation of Gut Microbiota by Cacao: Insights from an in vitro Model" contains the following issues.
Major issue:
The authors concluded that different enterotypes respond to cacao differently, despite the limited number of Prevotella-type samples, only 8 in this study. However, there was only a PCoA chart showing enterotype analysis results (Fig1A) without revealing the main contributors of each enterotype. It is also unclear whether this enterotype analysis is based on 0h of which group. On the other hand, when examining the microbiota composition in the supplement figure S1, the dominant genus in the control sample is usually Bacteroides, suggesting that these samples are likely of the Bacteroides type, not the Prevotella type. (I understand this is 24h, which differs from the 0h sample. However, it is difficult to believe that these samples are truly of the Prevotella type. If there were no Prevotella-type samples in this study, the conclusion in this manuscript can not be proved.
Minor issues:
Like the enterotype analysis, many details were missing either in the method section or the results. For example, in the "Gut Microbiota Data Analysis" paragraph, not only the approach but also the tools used in the analysis should be appropriately referenced. Even if they are in-house tools, the authors should clearly state them to demonstrate the study's reproducibility.
Additionally, the sentence "The beta diversity (weighted and unweighted UniFrac) metrics were estimated using Permutational multivariate analysis of variance (PERMANOVA)" is incorrect. PERMANOVA is not used to "estimate" UniFrac distance but to "assess" the beta diversity between groups.
Another concern in this study is whether it is appropriate to apply the cacao powder directly to the in vitro fecal incubation model to study the effects of whole cacao consumption. After the intake of cacao, the digested components may not be the same as the initial ones. The authors also mentioned that the study requires further in vivo validation.
Figures can be further improved. For example, the paragraph starting from line 258 discussed the Fig. 3D, E, but it isn't easy to follow which genus is at which part of the figure. Also, the purpose of the asterisk in Fig. 3D is unclear.
Author Response
# Reviewer1
Comments 1: Major issue: The authors concluded that different enterotypes respond to cacao differently, despite the limited number of Prevotella-type samples, only 8 in this study. However, there was only a PCoA chart showing enterotype analysis results (Fig1A) without revealing the main contributors of each enterotype. It is also unclear whether this enterotype analysis is based on 0h of which group. On the other hand, when examining the microbiota composition in the supplement figure S1, the dominant genus in the control sample is usually Bacteroides, suggesting that these samples are likely of the Bacteroides type, not the Prevotella type. (I understand this is 24h, which differs from the 0h sample. However, it is difficult to believe that these samples are truly of the Prevotella type. If there were no Prevotella-type samples in this study, the conclusion in this manuscript can not be proved.
Answer: We thank the reviewer for pointing out the need for greater clarity regarding our enterotype analysis. Gut microbiota enterotyping was performed based on the genus-level taxonomic profiles of fecal samples at 0 h (n = 48). The clustering results are presented in Figure 1A. To provide further detail, Table S1 summarizes the genus-level composition of each sample at 0 h (line 175), which served as the basis for enterotype classification. Based on these data, the samples were distinctly grouped into two enterotypes: Bacteroides type and Prevotella type.
Comments 2: Minor issues: Like the enterotype analysis, many details were missing either in the method section or the results. For example, in the "Gut Microbiota Data Analysis" paragraph, not only the approach but also the tools used in the analysis should be appropriately referenced. Even if they are in-house tools, the authors should clearly state them to demonstrate the study's reproducibility.
Additionally, the sentence "The beta diversity (weighted and unweighted UniFrac) metrics were estimated using Permutational multivariate analysis of variance (PERMANOVA)" is incorrect. PERMANOVA is not used to "estimate" UniFrac distance but to "assess" the beta diversity between groups.
Answer: We appreciate the reviewer’s helpful comments. To improve clarity and ensure reproducibility, we have revised the “Gut Microbiota Data Analysis” section to include detailed descriptions of the analytical approaches and tools used, including appropriate references (lines 119–140). In addition, we corrected the inaccurate statement regarding PERMANOVA.
(Lines 119–120) “The analysis of gut microbiota data was performed as escribed in previous studies (Kim et al. 2024a, Kim et al. 2024b).”
(Lines 125–127) “The beta diversity (weighted and unweighted UniFrac) metrics were used to assess differences between groups via Permutational multivariate analysis of variance (PER-MANOVA) using q2-diversity (Callahan et al. 2016).”
(Lines 135–140) “To identify gut microbiota enterotypes, baseline fecal samples were analyzed at the genus level. Clustering was conducted using the Partitioning Around Medoids (PAM) method based on Jensen–Shannon divergence (JSD) as the distance metric within the R environment (Arumugam et al. 2011). The optimal number of clusters and cluster robustness were determined by evaluating the Calinski–Harabasz (CH) index and silhouette analysis (Calinski and Harabasz. 1974).”
Comments 3: Another concern in this study is whether it is appropriate to apply the cacao powder directly to the in vitro fecal incubation model to study the effects of whole cacao consumption. After the intake of cacao, the digested components may not be the same as the initial ones. The authors also mentioned that the study requires further in vivo validation.
Answer: We have revised the Conclusion section to more explicitly acknowledge this limitation and emphasize the need for further in vivo validation (lines 342–345).
(Lines 342-345) “While this study used whole cacao powder in an in vitro fecal model, we acknowledge that digestion may alter its components in vivo. Further in vivo studies are required to validate these results and gain deeper insights into the impact of cacao on the gut microbiome.”
Comments 4: Figures can be further improved. For example, the paragraph starting from line 258 discussed the Fig. 3D, E, but it isn't easy to follow which genus is at which part of the figure. Also, the purpose of the asterisk in Fig. 3D is unclear.
Answer: We have updated the corresponding text in the Results section (lines 267–272) to follow the order in which the genera are presented in Figures 3D and 3E. Additionally, we clarified the meaning of the asterisk (*) and other symbols in the legends of Figures 1–3 (lines 192–193, 250–252, and 296–297).
(Lines 267–272) “In the Bacteroides enterotype, cacao treatment significantly increased the relative abundance of Lachnospiraceae NK4A136 group, Faecalibacterium, and Agathobacter, while decreasing that of Lachnospiraceae UCG 010 (Fig. 3D, E). In the Prevotella enterotype, cacao treatment significantly reduced the relative abundance of Lachnoclostridium, Bilophila, and Dorea, and increased the relative abundance of Prevotella (Fig. 3D, E, and S1).”
(Lines 192–193) “Significant differences are marked with an asterisk (*, p < 0.05). Significant increases in B type or P type are indicated in brown or blue, respectively.”
(Lines 250–252) “Significant differences between control and cacao groups are marked with an asterisk (*, p < 0.05). Significant increase in the control or cacao groups are indicated in brown or blue, respectively.”
(Lines 296–297) “Significant differences between control and cacao groups are marked with an asterisk (*, p < 0.05). Significant changes in B type or P type are indicated in brown or blue, respectively.”
Reviewer 2 Report
Comments and Suggestions for Authors
The manuscript was studied the impact of Cacao on gut microbiota using an in vitro fecal incubation model, considering enterotype-specific responses. It is useful and meaningful work for industry.
After carefully checking the manuscript, I could not recommend publication of manuscript at the current form.
- “Cacao is a rich source of proteins, lipids, minerals, starch, and pentosans and it is abundant in bioactive compounds, particularly polyphenols, which contribute to its antioxidant activity.” It is suggested that the specific content range be provided to facilitate the subsequent mechanism discussion.
- Figure 1A and C sampled 48 samples (A) but presented 46(C). Why were two samples deleted?
- As can be seen from Figure 2E, the degree of data dispersion is very large, and P is extremely less than 0.0001
- Mix all the data and then compare it with a single group. Why is that?
- The formatting of the references is inconsistent and very confusing throughout, please follow the journal formatting.
- Where is the analysis of the cacao mass spectrometry detection results? It is suggested that the author re-examine the materials and methods
。
Author Response
# Reviewer2
Comments 1: “Cacao is a rich source of proteins, lipids, minerals, starch, and pentosans and it is abundant in bioactive compounds, particularly polyphenols, which contribute to its antioxidant activity.” It is suggested that the specific content range be provided to facilitate the subsequent mechanism discussion.
Answer: We agree that providing specific content ranges can enhance mechanistic interpretation. However, the composition of cacao powder can vary depending on origin, processing methods, and analytical approaches. Although our mass spectrometry analysis does not provide precise quantitative content ranges, it offers valuable qualitative insight into the components present in the cacao powder used in our study. These results have now been included in the revised Supplementary Table S2 and are discussed in the manuscript (lines 223–239).
Comments 2: Figure 1A and C sampled 48 samples (A) but presented 46(C). Why were two samples deleted?
Answer: Two samples (Sub.17 and Sub.19) showed inconsistencies between their clustering results and genus-level compositions—Sub.17 clustered as Bacteroides type but had a high Prevotella abundance, and Sub.19 clustered as Prevotella type but showed low Prevotella abundance. To ensure classification accuracy, these samples were excluded from further analyses comparing the effect of cacao by enterotype. This decision is now clarified in the revised manuscript (lines 178–180), and detailed genus-level taxonomic compositions for each sample at 0 h are presented in Supplementary Table S1.
(Lines 178–180) “Two samples (Sub.17 and Sub.19) were excluded from further analysis due to their inconsistent Prevotella ratios, which could hinder accurate classification (Table S1).”
Comments 3: As can be seen from Figure 2E, the degree of data dispersion is very large, and P is extremely less than 0.0001
Answer: We thank the reviewer for pointing this out. Upon re-examination based on reviewer’s observation, we acknowledge that the data in Figure 2E indeed show a wide dispersion. However, the statistical difference between the control and cacao groups remains highly significant (P < 0.0001), confirming the robustness of our findings. We appreciate your careful review.
Comments 4 : Mix all the data and then compare it with a single group. Why is that?
Answer: Our study aimed to compare the effects of cacao based on enterotype. We first presented the enterotyping results at 0 h in Figure 1. Then, in Figure 2, we showed the overall impact of cacao treatment across all 24 h samples. Finally, in Figure 3, we performed stratified comparisons by enterotype. This stepwise analytical flow from general observations to enterotype-specific responses was designed to improve clarity and guide readers through the data interpretation.
Comments 5: The formatting of the references is inconsistent and very confusing throughout, please follow the journal formatting.
Answer: We appreciate the reviewer’s feedback. We have carefully reviewed and corrected all reference formatting to ensure full consistency with the journal’s guidelines.
Comments 6: Where is the analysis of the cacao mass spectrometry detection results? It is suggested that the author re-examine the materials and methods
Answer: The results of the mass spectrometry analysis of cacao powder components were originally included in Table S1. This information has now been updated and moved to the revised Supplementary Table S2, and is described in the Materials and Methods section (lines 142–156) of the manuscript.
Round 2
Reviewer 1 Report
Comments and Suggestions for Authors
The author addressed most of the issues, but not the most concerning part.
The supplementary table does show that there are 8 samples enriched with the Prevotella genus at 0 hours. However, the control samples becoming Bacteroides genus dominant is really weird (Fig. S1).
In addition, the number of Prevotella types is too small to support the conclusion. The author needs more samples and to reanalyze the abundance of the microbiota.
Author Response
# Reviewer 1
Comments 1: The author addressed most of the issues, but not the most concerning part. The supplementary table does show that there are 8 samples enriched with the Prevotella genus at 0 hours. However, the control samples becoming Bacteroides genus dominant is really weird (Fig. S1).
Answer: Thank you for your comment. We also observed that Bacteroides became dominant in both the control and cacao-treated groups after 24 hours of in vitro incubation, even among samples initially enriched with Prevotella. We believe this shift may be attributed to limitations inherent in the in vitro system, such as nutrient depletion, pH changes, or transient oxygen exposure, all of which may suppress the growth of Prevotella while favoring Bacteroides proliferation.
To address this issue, our analysis focused on the relative differences between the cacao-treated and non-treated control groups under identical incubation conditions. This approach enabled us to more reliably assess the effect of cacao, minimizing the confounding influence of incubation-associated shifts.
In addition, to improve clarity and avoid confusion, we have revised the labeling in Figure S1 to indicate the control group as "non-treated control." This change is intended to help readers clearly distinguish between baseline community structure and post-incubation changes.
For future studies, incorporating shorter incubation periods (e.g., 6 or 12 hours) or conducting in vivo validation may provide a more accurate reflection of microbiota dynamics in response to cacao treatment. We have added the following sentences to suggest the limitations of this study.
Line 273-, revised manuscript:
For future studies, incorporating shorter incubation periods (e.g., 6 or 12 hours) or in vivo validation may provide a more accurate reflection of microbiota dynamics in response to cacao treatment, while minimizing compositional shifts introduced by prolonged in vitro incubation.
Comments 2: In addition, the number of Prevotella types is too small to support the conclusion. The author needs more samples and to reanalyze the abundance of the microbiota.
Answer: We agree that the number of samples representing the Prevotella enterotype was limited in this study, which may constrain the generalizability of the findings for this group. However, all experiments were conducted simultaneously under identical controlled conditions to ensure consistency across samples. As such, it is not feasible to include additional samples at this stage.
In addition, all fecal samples were randomly collected from healthy volunteers, and we have recently observed a declining prevalence of the Prevotella enterotype in our collections. This trend may reflect broader dietary shifts, such as reduced fiber intake, which is known to be associated with lower Prevotella abundance.
Despite the limited number of Prevotella-type samples, we carefully performed statistical analyses to assess changes in microbiota composition and the effects of cacao treatment across all enterotypes. We have acknowledged the limitations and have taken a cautious approach in interpreting the results for the Prevotella group. We appreciate the reviewer’s valuable insight.
Line 345-, revised manuscript:
However, several limitations should be acknowledged. First, the number of individuals classified into the Prevotella enterotype was relatively small (n = 9), which may limit the generalizability of findings related to this group. Second, the in vitro fecal incubation model, while useful for screening, does not fully recapitulate host-related factors such as digestion, absorption, metabolism, and host–microbe immune interactions.
Reviewer 2 Report
Comments and Suggestions for Authors
accept
Comments on the Quality of English Language.
Author Response
We sincerely thank the reviewers for their valuable comments on our manuscript.
Round 3
Reviewer 1 Report
Comments and Suggestions for Authors
In the author's reply, "We also observed that Bacteroides became dominant in both the control and cacao-treated groups after 24 hours of in vitro incubation, even among samples initially enriched with Prevotella. We believe this shift may be attributed to limitations inherent in the in vitro system, such as nutrient depletion, pH changes, or transient oxygen exposure, all of which may suppress the growth of Prevotella while favoring Bacteroides proliferation."
This shift actually makes all your analyses unreliable. The in vitro system didn't seem to work well, and the analysis was based on suspicious data.
It is difficult to get your conclusion in the manuscript without redoing it or adding more supporting data.
Author Response
Comments 1: In the author's reply, "We also observed that Bacteroides became dominant in both the control and cacao-treated groups after 24 hours of in vitro incubation, even among samples initially enriched with Prevotella. We believe this shift may be attributed to limitations inherent in the in vitro system, such as nutrient depletion, pH changes, or transient oxygen exposure, all of which may suppress the growth of Prevotella while favoring Bacteroides proliferation."
This shift actually makes all your analyses unreliable. The in vitro system didn't seem to work well, and the analysis was based on suspicious data.
It is difficult to get your conclusion in the manuscript without redoing it or adding more supporting data.
Answer: We appreciate your comment and fully acknowledge the validity of your concerns. As you pointed out, the small sample size in the Prevotella group and the increased dominance of Bacteroides after 24 hours of in vitro incubation are important limitations that may influence the interpretation of our findings. To address these concerns, we have revised the manuscript to reduce the emphasis on enterotype-specific results—particularly those involving the Prevotella group—and we now describe these findings with greater caution. Accordingly, we have updated the abstract, discussion, and conclusion to reflect these changes. In addition, the manuscript title has been revised so that enterotype is no longer presented as a central focus of the study.
Although it is not feasible to repeat the experiment at this stage, we believe the revised discussion and clearer explanation of limitations help address the concerns raised.
Furthermore, this study was conducted using fecal samples from randomly recruited participants, and the results underscore that individual variability in gut microbiome composition can lead to divergent responses to the same dietary intervention. While the limited number of Prevotella-dominant participants is indeed a constraint, we believe that our findings still contribute meaningfully by highlighting the need for future studies with more balanced enterotype representation. Specifically, our results point to the importance of carefully designed follow-up studies that recruit participants in proportion to each enterotype to more rigorously evaluate diet–microbiota interactions.
Revised Manuscript:
Title “Modulation of Gut Microbiota by Cacao: Insights from an in vitro Model”
Lines 19-21 “This study investigated the impact of cacao on gut microbiota using an in vitro fecal incubation model with samples from healthy Korean adults.”
Lines 24-27 “Conversely, in the Prevotella enterotype, cacao treatment was associated with an increase in the relative abundance of Prevotella; however, the small sample size and community shifts during incubation limit the robustness of this observation.”
Lines 31-33 “These findings suggest a potential role for cacao as a gut microbiome modulator, highlighting its possible utility in microbiome-targeted dietary interventions and therapeutic strategies.”
Lines 270-275 “In the Prevotella enterotype, cacao treatment was associated with a reduction in the relative abundance of Lachnoclostridium, Bilophila, and Dorea, alongside an increase in the relative abundance of Prevotella (Fig. 3D, E, and S1). These results may suggest that cacao contributes to the maintenance of Prevotella populations, particularly in Prevotella-dominant microbiota in our dataset.”
Lines 291-293 “This trend suggests that cacao may promote the growth of beneficial Bifidobacterium species, which are known to support gut health. Further studies are needed to confirm these effects in vivo and clarify individual variability.”
Line 306 “Predictive KEGG Functional Profiling According to Cacao Treatment”
Lines 329-333 “In the Prevotella enterotype, cacao treatment was associated with an increased representation of pantothenate/CoA biosynthesis, and a decreased representation of methane metabolism (LDA > 2.0, Table 1). P. copri, a species that showed increased relative abundance following cacao treatment, possesses a biosynthesis pathway for vitamin B5 (pantothenate) [50], suggesting a potential link between its enrichment and increased production of vitamin B5.”
Lines 339-356 “Our findings suggest that cacao has the potential to modulate the composition and functional profile of the human gut microbiota. Using an in vitro fecal incubation model with samples from 48 healthy Korean adults, we observed significant shifts in microbial community structure and predicted metabolic pathways following cacao treatment. These alterations included increases in health-associated bacterial genera and enhancement of pathways related to carbohydrate and vitamin metabolism. While some variation in response appeared to be associated with enterotype, this observation should be interpreted with caution due to several limitations. First, the number of individuals classified into the Prevotella enterotype was relatively small (n = 9), which may limit the generalizability of findings related to this group. Second, the in vitro fecal incubation model, while useful for screening, does not fully recapitulate host-related factors such as digestion, absorption, metabolism, and host–microbe immune interactions. In addition, a shift to-ward Bacteroides dominance was observed in most samples after 24 h of incubation, including those initially enriched with Prevotella, indicating that the in vitro system may not fully preserve baseline microbial community structures. While this study used whole cacao powder in an in vitro fecal model, we acknowledge that digestion may alter its components in vivo. Further in vivo studies are required to validate these results and gain deeper insights into the impact of cacao on the gut microbiome.”

Round 4
Reviewer 1 Report
Comments and Suggestions for Authors
I can understand the difficulties authors face in adding experimental data.
Since the author has toned down the statements in the manuscript that are not sufficiently supported by the data at this time, there are no further comments.